# Are Food Hubs Sustainable? An Analysis of Social and Environmental Objectives of U.S. Food Hubs

Haniyeh Shariatmadary [1,*], Sabine O'Hara [1], Rebecca Graham [1] and Marian Stuiver [2]

[1] College of Agriculture, Urban Sustainability and Environmental Sciences (CAUSES), University of the District of Columbia (UDC), 4200 Connecticut Ave. NW., Washington, DC 20008, USA

[2] Green Cities Programme, Wageningen University and Research (WUR), 6708 PB Wageningen, The Netherlands

\* Correspondence: haniyeh.shariatmadar@udc.edu

**Abstract:** The United States food system is highly centralized with only three of the fifty states producing more than 75 percent of U.S. fruits and vegetables. The high reliance on long-distance transportation and cold chains undermines the sustainability of the food system and adds to its vulnerability. This was most recently demonstrated during the COVID-19 pandemic which caused significant disruptions to food supply chains. A promising alternative is a more decentralized and localized food system which reduces the reliance on long-distance transportation and long supply chains. Since such a food system will likely consist of smaller producers, questions have been raised about its economic viability. This precipitated the idea of Food Hubs as market aggregators. The model was first introduced by the U.S. Department of Agriculture as a way to aggregate the agricultural product of small farms. It has since evolved to imply a more flexible food system that can complement various parts of the food supply chain. This study develops a framework to assess the social and environmental sustainability contributions of Food Hubs and especially of urban Food Hubs, since 80 percent of U.S. food consumers live in urban and metro areas. Using our framework, we conducted a content analysis of publicly available information for 50 Food Hubs in metropolitan areas across the United States. We find that Food Hubs contribute to environmental sustainability by reducing food transportation through sourcing from local farms. They also perform relatively well in contributing to lowering food waste and loss. Their contributions to improving water management and adopting more sustainable food production methods, however, appear to be less strong. Similarly, Food Hubs appear to enhance some of our selected aspects of social sustainability such as improving access to fresh and healthy food to local consumers, and organizations such as schools and hospitals. Only a few of the Food Hubs in our sample, however, address our other aspects of social sustainability such as improving food security. We conclude our study by offering an aggregate ranking of the sustainability contributions of our selected Food Hubs based on our assessment framework.

**Keywords:** Food Hubs; sustainable food systems; food security; social and environmental sustainability

## 1. Introduction

The United States food system is highly centralized and driven by economies of scale. This is true not only for commodity crops but also for specialty crops focused on human consumption. According to the U.S. Department of Agriculture, specialty crops include fruits, vegetables, herbs, nuts, and other agricultural products. The high degree of centralization of U.S. agriculture results in a substantial reliance on long-distance transportation and cold chains which undermines the sustainability of the food system and adds to its vulnerability. This suggests that the U.S. food system achieves its low food prices by leaving negative social and environmental externalities unaccounted for.

In this paper, we explored the social and environmental externalities of the U.S. food system by examining the sustainability of Food Hubs, which have been proposed as a more decentralized and localized alternative to the current highly centralized food system. U.S. Food Hubs have emerged in cities and metropolitan areas across the country over the past thirty years. They were intended as market aggregators to improve the position of small producers in the marketplace. We focus our analysis by developing a sustainability framework that draws on the shortcomings of the current centralized food system of the United States. Using publicly available information, we rate U.S. Food Hubs located in highly populated urban and metropolitan areas and assess the strengths and weaknesses of the Food Hub based on publicly available information.

A review of the current food system of the United States illustrates that agricultural production contributes little to the food supply most consumers would consider agricultural products. For example, corn and soybeans comprise more than half of all harvested cropland, and 40 percent of this harvest goes to animal feed and ethanol production [1]. Only a small percentage of agricultural production consists of so-called specialty crops, which the United States Department of Agriculture (USDA) defines as "Fruits and vegetables, tree nuts, dried fruits, horticulture, and nursery crops (including floriculture)." According to the USDA, Specialty Crops are used for human food consumption, but also for medicinal purposes, and/or aesthetic gratification [2]. Therefore, we use the terms "specialty crops", "fruits and vegetables", and "food" interchangeably, implying the more direct use of agricultural products for human consumption.

Specialty crops account for only three percent of all cropland in the United States [3]. However, even the small percentage of land allocated to specialty crop production is highly centralized. More than 75 percent of the vegetables grown in the United States stem from three states: California (57 percent); Arizona (12 percent); and Florida (8 percent) [4,5]. It is noteworthy that two of these states, namely California and Arizona, are rapidly running out of water.

A highly centralized food system may supply food at low costs by utilizing economies of scale, but it performs poorly in meeting social and environmental sustainability criteria [6]. It is also more vulnerable to disruptions, as evident during the recent COVID-19 pandemic. From an environmental perspective, the energy consumption and air pollution impacts associated with the long-distance transportation needs of a centralized food system are problematic, especially in light of rising energy prices [7]. The need for extensive cold chains and storage further adds to the environmental burden [8]. From a social perspective, a highly centralized food system disconnects food production from food consumption and commercializes food. In a commercialized food system, economic motives play the dominant role while non-economic values including quality, disparities in access, the preservation of landscapes, and other less tangible characteristics such as the connection to nature become less important. A USDA initiative called 'know your farmer know your food' has tried to address some of these disconnects in the U.S. food system [9]. Recent studies have also pointed to the vulnerability of the U.S. food system experienced during the COVID-19 pandemic. These findings do not speak well for the ability of the current U.S. food system to deal with future shock events [10–12]. In addition, climate change is placing pressure on some of the key specialty crops producing states in the western U.S., which has experienced severe droughts in recent years [13–16].

Given the limited focus of U.S. agriculture on specialty crops, the country relies heavily on specialty crops imports, which in turn creates a significant carbon footprint. In 2015, U.S. exports of specialty crops totaled USD 26.4 billion, while imports totaled USD $48.9 billion leaving a trade deficit of USD 22.5 billion. This trade deficit has widened steadily [17] with the majority of fruit and vegetable imports coming from Mexico (44 percent), Canada (12 percent), and Chile (8 percent) [18].

Concerns have also been raised about the food quality implications of a highly centralized food system [19,20]. Many specialty crops suffer nutrient losses during long transportation via air, rail, and road resulting in either a lower nutrient density or outright food

losses [21]. Long food supply chains also rely heavily on food preservation methods which may have negative food quality and environmental impacts [22].

A possible alternative to a highly centralized food system is a more decentralized, localized one. Several studies have argued the advantages of such a localized food system [15,23,24]. They include reduced transportation and cold chain needs as food is produced closer to where most consumers live. In addition, a localized food system can also add social value as food consumers can forge new connections to their food including a closer connection with nature [25–27].

Definitions of what constitutes a local food system vary. In the United Kingdom, for example, a geographical distance of 30–40 miles is considered local, with the exception of the London metropolitan area, which considers food from within 100 miles as local. Canada also refers to a local diet as a "100-mile diet" [28,29], whilst Washington, D.C., considers food from within a 150-mile radius as local [30], and according to the U.S. Congress, up to 400 miles is considered local [31]. A common notion is that a local food system grows and distributes food more locally through direct sales to consumers or "the unification of food production and consumption within the same physical and social space" [32,33]. Given the current pressures on global food supply chains caused by external shocks associated with climate change and rising energy costs, the advantages of a more distributed, localized food system have become even more apparent [34].

However, localized food systems have typically been considered less viable from an economic perspective [25]. Small producers may not be able to compete and may lack access to a large enough share of the consumer market to produce sufficient revenue [35]. As a result, the concept of the Food Hub was proposed to act as a market aggregator that assists small producers in gaining a more robust market share for their products [36]. According to the USDA, a food hub is "a centrally located facility with a business management structure facilitating the aggregation, storage, processing, distribution, and/or marketing of locally/regionally produced food products (Figure 1) [24,37]. More expanded aspects of the model also include networking and educational components for both small producers and consumers (see Figure 1) [37]. Food Hubs can, therefore, be considered an important feature of a more decentralized, localized food system that can alter or at least complement the current food supply and mitigate the risks associated with its long supply chains [38]. As consumers increasingly want to know where their food comes from, Food Hubs may also play an important role in providing better information and increased transparency about where and how food is produced and handled throughout the entire food supply chain [29,39].

Despite the considerable attention Food Hubs have received as a potentially resilient and sustainable food production alternative, only a few studies have explored how Food Hubs promote sustainability at a country level in practice [40,41]. The need for more reliable databases is likely the main reason for the gap in the literature. Moreover, Food Hubs in the United States constitute an exceedingly small fraction of total food production [42]. The few existing studies of U.S. Food Hubs have primarily focused on the economic and operational aspects of Food Hubs with little attention given to their social and environmental contributions [43]. Our study takes a step toward filling this information gap.

In building on the general literature on social and environmental sustainability, we build on the discussion of elements of sustainability by developing a hierarchical framework that captures these main elements and systematizes them. Our framework thus facilitates the evaluation of a food production model and its contributions to social and environmental sustainability. Using this novel framework and in the absence of a reliable database that captures social and environmental impacts, we use publicly available information and rank the contributions of Food Hubs to specific aspects of social and environmental sustainability.

We focus our study on Food Hubs in urban and metropolitan areas in the United States since this is where the vast majority of U.S. food consumers live. According to the 2019 U.S. Census, 83 percent of the U.S. population lives in cities and metro areas [44].

Our sample of 50 Food Hubs starts with some of the most populous urban areas since Food Hubs can play an especially important role in these areas in reducing the impact of long supply chains and improving food access. The growing demand of U.S. consumers for locally produced specialty crops adds further relevance to our study areas [45,46].

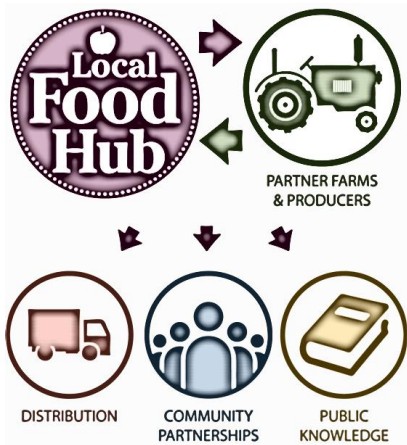

**Figure 1.** Food Hubs can play a key role in filling gaps in the local food supply chain. They partner with growers and may offer a range of services, including aggregating, marketing, selling, and processing local food, and educating farmers and consumers on local food benefits and best practices.

## 2. Methodology—Developing a Sustainability Framework

The main objective of our study is to evaluate the contributions that Food Hubs make to the social and environmental sustainability of the U.S. food system. We base our analysis on publicly available information to assess whether the Food Hubs themselves aim to achieve social and environmental sustainability goals. Our framework is based on a three-pronged approach. First, we develop indicators that capture key elements of social and environmental sustainability. We identify these indicators of social and environmental sustainability by reviewing the literature on food system sustainability. Secondly, we use our indicators to analyze relevant information on the social and environmental sustainability contributions of 50 Food Hubs located in cities and metropolitan areas across the United States based on publicly available information. We take this approach since there is no established database that tracks the social and environmental sustainability of U.S. Food Hubs. In fact, very limited information exists about the estimated 400 U.S. Food Hubs, although efforts are currently underway by the USDA to improve data availability. Third, to facilitate our analysis, we establish sub-indicators within our social and environmental sustainability criteria. We chose these sub-indicators to facilitate our analysis based on our initial review of the publicly available information about Food Hubs. Finally, we develop a four-step rating system to evaluate the performance of Food Hubs in our sample based on their contributions to our selected sub-indicators. We further summarize the ratings based on our sub-indicators across our sample of Food Hubs to construct an average achievement rate for each sub-indicator. This provides an aggregate achievement rating of all Food Hubs in our sample based on their social and environmental sustainability contributions.

### 2.1. A Framework to Assess the Sustainability of U.S. Food Hubs

Efforts to move towards a more sustainable food system have been largely driven by concerns about the negative environmental and social externalities associated with the current food system [47–49]. According to the Food and Agriculture Organization (FAO) of the United Nations, food system sustainability is also linked to food and nutrition security, whereby a sustainable food system is defined as "… a food system that delivers food security and nutrition for all in such a way that the economic, social and

environmental bases to generate food security and nutrition for future generations are not compromised" [7,50]. A sustainable food system thus implies that it provides reliable access to nutritious food at every location and over time.

Drawing on the literature on sustainable food systems, we develop a framework for assessing the social and environmental sustainability of Food Hubs. The goal of this approach is to evaluate the viability of Food Hubs as a more sustainable decentralized alternative to the current food system. We then apply this framework, which is summarized in Figure 2, to our 50 selected Food Hubs in some of the most populous urban and metro areas across the U.S.

Social Sustainability. We identify two indicators to capture the social sustainability contributions of the Food Hubs in our sample, namely (1) their contribution to 'improved access to fresh and healthy food' and (2) their contribution to 'improved food security'. The two indicators are related, however, they are not the same. Improved food access might be considered an essential first step, yet improved access does not guarantee improved food security.

Access to Fresh and Healthy Fruits and Vegetables. Lack of access to affordable, healthy food clearly has implications for health and well-being [51]. Households who lack access to affordable healthy food have higher incidents of food-related illnesses such as diabetes, obesity, and heart disease [52,53]. The social and political determinants of low food access are well documented and include income, race, and ethnicity [54]. Sadly, the health implications of unequal food access tend to further amplify social disparities as poor health impacts the capacity to generate a living wage and limits the formation of social capital [55]. However, access alone, does not guarantee that the food will actually be consumed and that the disparities resulting from poor food access are addressed.

Addressing Food Insecurity. Our second social sustainability indicator is related to food access, however, it adds further complexity. One of the tools the USDA utilizes to assess the degree of food insecurity among U.S. households is based on responses to an annual food security survey. Survey questions ask respondents to rank the degree to which statements such as "I worried whether our food would run out before we got money to buy more" apply to them. Another survey question asks respondents, "Did a child in the household ever not eat for a full day because you couldn't afford enough food?" [56]. According to the 2021 Current Population Survey, 10.2 percent of U.S. households are food insecure [57]. Students at U.S. colleges and universities are a subset of consumers that appears to experience particularly high levels of food insecurity. Among college students, 41 percent reported that they were food insecure at least during part of the previous year [44]. The health impacts of food insecurity are well documented and include adverse physical and mental health effects [52,58,59]. Recent studies suggest that food insecurity worsened during the COVID pandemic when school-aged children did not receive school meals [60]. Others suggested that the impact was inconclusive, given the state- and national-level initiatives launched to address food insecurity and supply chain disruptions during the pandemic. Despite the challenges of capturing the complexities of food security, it is a generally accepted indicator of social impacts, and we use it here as our second indicator to assess the social sustainability contributions of Food Hubs.

Environmental Sustainability. Our four selected environmental indicators also build on the literature and the reported negative environmental externalities of the current food system and its centralization. They are: (1) reduced transportation; (2) more healthy and sustainable food production practices; (3) reduced food waste and loss; and (4) improved water management (see Figure 2).

Reducing Food Transportation. We argue that a more decentralized and localized food system reduces the need for long-distance transportation and storage [61]. Research estimates that 14 percent of U.S. energy consumption and 38 percent of U.S. greenhouse gas emissions are attributable to food transportation [50,62]. By shortening the distance between food production and consumption, these negative externalities can be reduced.

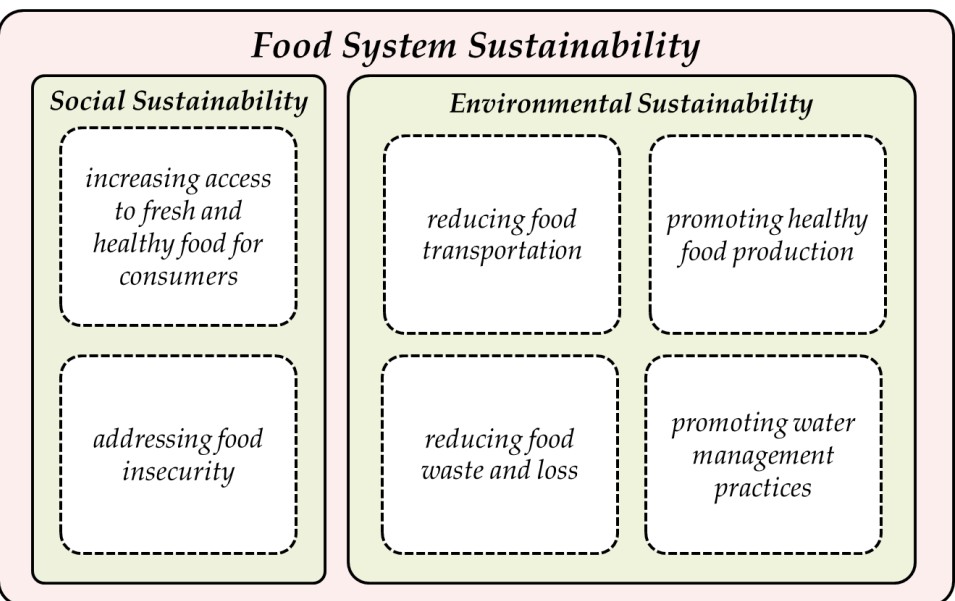

**Figure 2.** A sustainability framework of selected social and environmental indicators identified in the literature as addressing important aspects of sustainability.

Promoting Healthy Agricultural Practices (Food Production). We further argue that more sustainable food production practices including the reduced use of pesticides, and practices that reduce water pollution and soil degradation improve the environmental sustainability of food production [63]. Conventional U.S. agriculture relies heavily on the use of pesticides and used 1.2 billion pounds in 2016. More than 500 active pesticide ingredients have been approved in the U.S. since the 1970s. Among them are 72 pesticide ingredients that have been banned in the European Union (EU). This equates to 27 percent of all pesticides in use in the U.S. being banned in the EU [64]. The indicator 'more healthy and sustainable food production' assesses whether a Food Hub practices and/or promotes agricultural production methods that use less pesticides and implement more sustainable crop management practices.

Reducing Food Waste. Given the long food supply chains associated with a highly centralized food system, a significant share of fruits and vegetables produced is wasted at various stages of the supply chain, including on the farm, during storage, transportation, and distribution, as well as during the food processing, and consumption stage. An estimated 30–40 percent of the U.S. food supply is wasted across the supply chain. Our third indicator of the environmental sustainability contributions of Food Hubs is, therefore, their contribution to reducing food waste and loss [65].

Promoting Water Management Practices. This leads us to our fourth indicator, namely improved water management. Crop production, and especially the production for specialty crops (fruits, vegetables, herbs, nuts, etc.) requires enormous amounts of fresh water. Globally, agriculture appropriates 75 percent of freshwater use, and given the high percentage of food waste in the U.S., an estimated 25 percent of freshwater and 300 million barrels of oil are wasted annually in the form of water and petroleum products embodied in the food waste and loss generated across the food supply chain [66]. According to the National Water Quality Assessment project, agricultural runoff is also the leading cause of water pollution in rivers and streams, the second largest source of threats to wetlands, and the third largest polluter of lakes [67]. We capture these impacts in the indicator 'promoting water management practices'.

In addition to being based on our review of the literature on sustainable food systems, our sustainability indicators are also well aligned with the Sustainable Development Goals (SDGs) of the United Nations. Our indicators 'Increasing access to healthy food' and 'promoting healthy food production' reflect SDG 3 'good health and well-being', and

SDG 12 'responsible consumption and production'. Our second indicator of social sustainability, 'addressing food insecurity', is closely related to the SDG 2 'zero hunger', and SDG 10 'reduced inequality'. Our environmental sustainability indicators 'reducing food /transportation', 'reducing food waste', and 'promoting water management' are well aligned with SDG 13, 'climate action', SDG 15 'life on land', and SDG 16 'clean water and sanitation'. Given our focus on cities and metropolitan areas as a basis for selecting the Food Hubs in our sample, our study is also aligned with SDG 17 'sustainable cities and communities'.

### 2.2. Sample Selection, Data Sources, and Analytical Approach

Data on U.S. Food Hubs are scarce, and existing data sources are focused almost exclusively on the economic viability of Food Hubs as market aggregators for small producers. To address the lack of data availability, we use publicly available information provided by the Food Hubs themselves to evaluate their contributions toward our six sustainability indicators. This approach of querying publicly available information to analyze the information content when data availability is limited has been applied in a number of fields including organizational studies, communication, political sciences, and the food industry [68–72]. Regrettably, the depth of available information varies across Food Hubs. A low ranking in a specific indicator category may therefore imply little publicly available information rather than a lack of contribution to the sustainability objectives captured by the indicator. For example, a Food Hub may post information about specific activities, such as composting food waste, on their websites as a means of communicating with their customers or farmers even though the goal of reducing food waste may not be mentioned in their mission and goal statement. We carefully choose our sub-indicators to ensure that we consistently captured the available information for the Food Hubs in our sample. While we recognize the limitations of basing our analysis on publicly accessible information, our analysis offers new insights into the sustainability contributions of Food Hubs that has previously remained unexplored. In addition, our analysis points to potential communication gaps and divergent objectives reflected in the activities of a Food Hub and their expressed mission and goals.

To choose our sample, we use the Food Hubs directory of the USDA [73]. The directory provides some basic information about the business model and activities of the included Food Hubs. However, regular updates of the database have been sparse. The biennial National Food Hub Survey conducted by Michigan State University and the Wallace Center takes a similar approach and mainly focuses on the business aspects of the Food Hubs included in the survey [74]. The directory includes 230 Food Hubs, which constitutes more than half of the estimated 400 Food Hubs in the United States [75].

In selecting our sample, we first excluded those Food Hubs that do not supply fruits and vegetables and focus only on those Food Hubs that predominantly supply specialty crops. We argue that these Food Hubs more likely meet our social sustainability goals of improved food access and food security. Next, we identify 50 Food Hubs in some of the most densely populated counties in the U.S. by matching the locations of the Food Hubs with the population density of counties from the 2020 U.S. Census. County population density is reported in persons per square kilometers. Population density in our sample ranges from under 500 to 13,000 persons per square kilometer, which corresponds to the 96th to 99.9th percentiles of all U.S. counties. Despite this relatively narrow range, our sample captures the equivalent of more than ten percent of the U.S. population. We chose this approach to analyze those Food Hubs which serve the largest share of food consumers (see Figure 3).

We use three sources to obtain information regarding the sustainability contributions of our Food Hubs sample: first, we use online information from the websites of the Food Hubs as well as from published reports, news updates and other publicly available materials; secondly, we use entries from the USDA Food Hubs directory; third, we use census

data to obtain socio-economic information by matching the county and zip code where a Food Hub operates with the available census data.

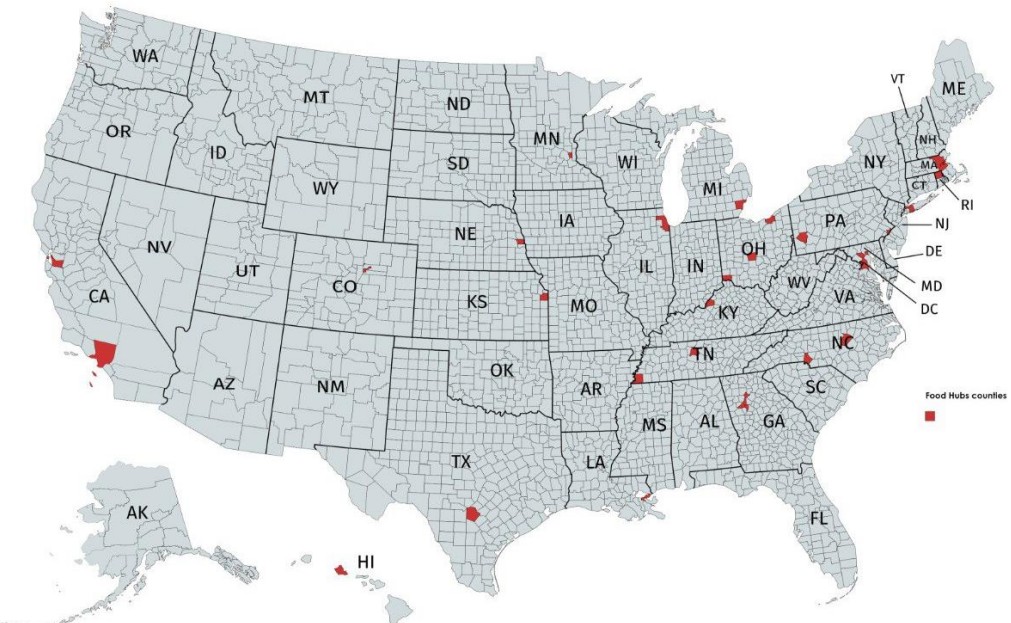

**Figure 3.** The 50 food hubs in the sample cover most populous urban areas in the US. Created with www.mapchart.net

### 2.3. Sub-Indicators and Rating Scale

Our initial review of publicly available information about the Food Hubs in our sample resulted in a further adjustment of our sustainability indicators in order to more fully capture the available information. We further break down each of the six sustainability indicators in our framework into sub-indicators of sustainability as shown in Tables A1 and A2 in Appendix A. For the two social sustainability indicators, we introduce four and two subcategories, respectively. For the four environmental sustainability indicators, we introduce two (health production methods) subcategories, four (transportation) subcategories, two (food loss and waste) subcategories, and one (water management) subcategory. In addition, we develop a ranking system that allows us to assess the performance of each Food Hub with respect to each indicator category based on a scale of 0–3.

As indicated in Tables A1 and A2, each of our subcategories is designed to query the available information and to establish whether a Food Hub specifically states their contribution to a social or environmental sustainability indicator. In our rating schedule, three represents a strong contribution expressed by the explicit mention of the indicator and its emphasis in the publicly available documents; 2 implies some contribution expressed by the explicit mention of the indicator; 1 implies a weak contribution expressed by a lack of explicit mention of the indicator but its implied recognition in the available documents; and 0 implies no contribution to a specific sustainability indicator category expressed by its absence in the available documents. For our selected sub-indicators that have a binary response, a rating of 3 implies a positive contribution, and 0 means no contribution.

Our analysis of the available information further indicates that some Food Hubs may amplify their impact by working with retailers. Some others sell not only to consumers, but also to local food businesses, food processors, and non-profits including schools and healthcare providers. Some serve as market aggregators for small producers, while others also act themselves as producers and educators. Depending on their organizational mission, Food Hubs may therefore make both direct and indirect contributions to our social and environmental sustainability indicators. For example, a Food Hub may increase access to fresh and healthy food by directly supplying fresh produce to local consumers, but

they may also offer classes about healthy food preparation and eating habits. These classes can be considered an indirect impact since they promote healthier food preparation and eating habits.

Similarly, some Food Hubs aggregate the produce of local farmers and take over transportation, marketing, and sales responsibilities for the farmers. Others also provide training and technical assistance to farmers to improve their productivity, reduce or recover waste, or teach water capture and reuse techniques. This kind of technical assistance can be considered a contribution to our environmental sustainability indicators 'waste reduction' and 'water management'.

Transportation may also have social implications in addition to its environmental implications since a lack of access and costly transportation may be barriers to food access for low-income consumers. To evaluate whether a Food Hub can meaningfully reduce food insecurity, we analyze the census data of the geographic area where a Food Hub is located and identifying the poverty rate of the county by consulting the 2019 Census' Small Area Income and Poverty Estimates (SAIPEs) [76]. To assess the transportation impact of the food supply side of a Food Hub rather than its food demand impact, we identify the number of farms with whom a Food Hub works. We assume that a larger number of member farms of a Food Hub will reduce the need for additional supplemental food transportation from a longer distance. Data limitations do not allow us to account for the size of the member farms of a Food Hub. However, the number of farms working with a Food Hub can be considered a proxy for the reach of a Food Hub.

### 2.4. Summarizing Rating Results

We present the results of our analysis in Table 1. The first four columns indicate the ratings that all Food Hubs collectively achieved in our sub-indicator categories based on our analysis of the publicly available information about each Food Hub. To complement this presentation of results in our four rating categories, we calculate an average achievement rate which is presented in the fifth column in Table 1. This average contribution of all Food Hubs to each of our sustainability indicators is calculated by assigning a value of 100 percent when all food hubs achieve a perfect score of 3 and 0 percent when they make no contribution. Assuming a rating scale of $n = 0, \dots, N$ and associated ratings of $\omega_0, \dots, \omega_N$, which add up to 1, i.e., $\sum_{n=0}^{N} \omega_n = 1$, we calculate the average achievement rate:

$$Average\ Achievement\ Rate = \frac{\sum_{n=0}^{N} n \times \omega_n}{N}$$

Assuming that all in our sample received a score of 0, then $\omega_0 = 1$ and $\omega_1, \dots, \omega_N = 0$, implying that the average achievement rate is 0. On the other hand, when all receive a perfect score, then $\omega_N = 1$ and $\omega_0, \dots, \omega_{N-1} = 0$, implying that the average achievement rate is 1. Our average achievement rate thus measured the contribution of all Food Hubs in the aggregate to our various sustainability indicators as falling between 0 and 1 with 0 being the lowest rating and 1 being a perfect score.

Figure 4 presents a higher-level aggregation to show the collective contribution of all Food Hubs for our indicator categories. These calculations are based on the averages of achievement rates in each of the sub-indicators presented in Table 1. In other words, we do not weight different indicator categories differently and do not assume a lower or higher contribution to sustainability associated with the respective contributions of the indicators to the overall sustainability of the Food Hubs in our sample.

**Table 1.** Results of Rating Food Hubs' Contributions to Social and Environmental Sustainability.

| 0 | 1 | 2 | 3 |
| --- | --- | --- | --- |

| No Contribution | Weak Contribution | Some Contribution | Strong Contribution | Average Achievement Rate |
|---|---|---|---|---|
| **Increasing Access to Fresh and Healthy Food** | | | | |
| Does the Food Hub state a social mission for increasing access to healthy and fresh food for local consumers? | | | | |
| 2% | 22% | 26% | 50% | **75%** |
| Does the Food Hub supply food to local K-12 schools, hospitals, or senior care? | | | | |
| 54% | | | 46% | **46%** |
| Does the Food Hub offer convenient and effective means for the direct sale of products to consumers? | | | | |
| 20% | 40% | 24% | 18% | **80% \*** |
| Does the Food Hub offer training programs to consumers to promote healthy choice and use of food? | | | | |
| 40% | | | 60% | **60%** |
| **Addressing Food Insecurity** | | | | |
| Does the Food Hub state a social mission to solve food insecurity in the region where it operates? | | | | |
| 52% | 10% | 18% | 20% | **35%** |
| Is the Food Hub located within reasonable distance from a low-income neighborhood? | | | | |
| 30% | 12% | 24% | 34% | **53%** |
| **Promoting Healthy Food Production** | | | | |
| Does the Food Hub require or highly encourage healthy food production practices (organic production, chemical-free production, or good agricultural practices (GAPs))? | | | | |
| 43% | | | 57% | **57%** |
| Does the Food Hub offer technical support or training programs to farmers about any aspect of food production? | | | | |
| 74% | 0% | 0% | 26% | **26%** |
| **Reducing Food Transportation** | | | | |
| Does the Food Hub have a large membership? | | | | |
| 8% | 26% | 26% | 40% | **65%** |
| Does the Food Hub procure from local farmers? | | | | |
| 0% | 24% | 12% | 64% | **80%** |
| Does the Food Hub offer home delivery, CSA, or neighborhood drop-off? | | | | |
| 36% | | | 64% | **63%** |
| Does the Food Hub sell products to local food businesses, including local large supermarkets, small grocery stores, corner stores, food retailers, restaurants, caterers, distributers, and food processors? | | | | |
| 40% | 34% | 20% | 6% | **31%** |
| **Reducing Food Waste and Loss** | | | | |
| Does the Food Hub donate unsold products? | | | | |
| 34% | 0% | 0% | 66% | **67%** |
| Does the Food Hub actively compost waste produce? | | | | |
| 88% | 0% | 0% | 12% | **13%** |
| **Promoting Water Management** | | | | |
| Does the Food Hub encourage water-saving production? | | | | |
| 96% | | | 4% | **1%** |

Note: The last column "Average Achievement Rate" shows the weighted average of ratings normalized to be a fraction of 100. A perfect score of 100% implies that all Food Hubs would receive a rating of 3. \* For this sub-indicator, we calculate the achievement rate based on the sum of 1–3, treating ratings 1–3 as a perfect score due to the binary nature of the sub-indicator.

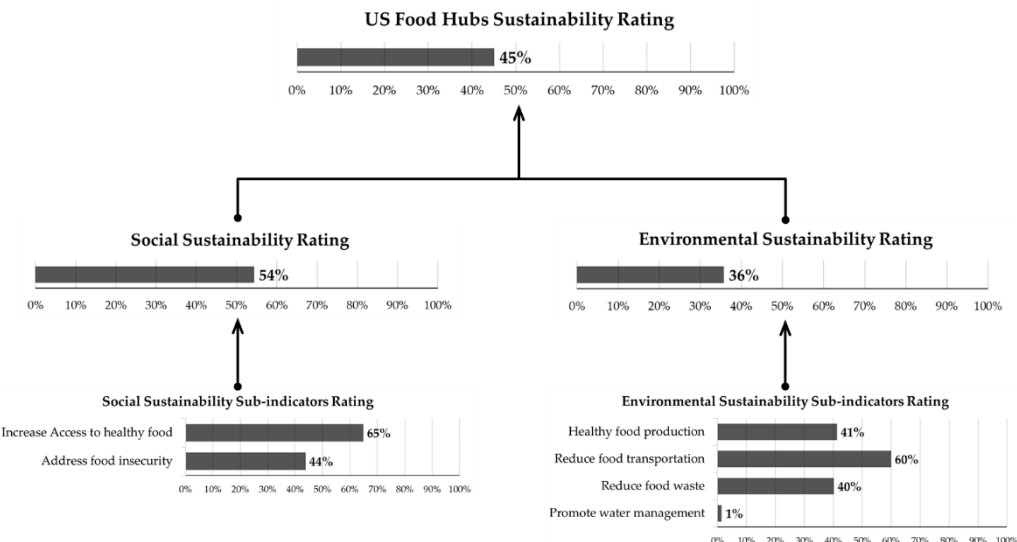

**Figure 4.** Aggregate Sustainability Rating of selected U.S. Food Hubs based on the social and environmental sustainability framework.

## 3. Discussion of Results

As previously discussed, the available data about Food Hubs in the United States do not currently allow a ready assessment of their sustainability contributions. Our analysis of publicly available information about 50 Food Hubs in urban and metropolitan areas across the U.S. suggests, however, that a rating system could be developed and would provide meaningful information about current practices as well as viable improvements. Our proposed sustainability indicators address both social and environmental sustainability contributions and offer a benchmark for the range of sustainability contributions of the U.S. Food Hubs in our sample. A high rating indicates a strong contribution of a Food Hub to a specific sustainability indicator, while a low rating would suggest a lack of contribution to a specific sustainability indicator.

The results of our analysis are presented in Table 1, which summarize the social and environmental contributions of the Food Hubs in our sample, respectively. We present the frequencies of the contribution to different aspects of social and environmental sustainability under each rating. For example, 50 percent of Food Hubs in our sample were rated as '3' for the first question in Table 1, which implies that they contributed to the sustainability category captured in question 1. The last column displays the average achievement rate as defined in the previous section. For example, Food Hubs in our sample could, on average, achieve 75 percent of the sustainability sub-indicator captured by the first question in Table 1.

In analyzing the social sustainability indicators and their subcategories, we find that the Food Hubs in our sample make a relatively consistent contribution to the objective of 'improved access to fresh, healthy produce'. Food Hubs thus have the potential to meaningfully contribute to the increased availability of fresh, locally produced food for local consumers. Some of the Food Hubs in our sample also supply food to specific sub-cohorts of consumers such as children in K-12 schools, hospital patients and staff, and residents in senior care facilities, where food quality may be of particular importance. Our findings further indicate that 80 percent of our Food Hubs offer at least some option of improving the convenience of accessing fresh, locally produced food through farmers markets, online sales, or community-supported agriculture (CSA). In addition, a number of the Food Hubs in our sample make an indirect contribution to 'improved food access' by offering programs to educate consumers on healthy diets and more informed food choices. While some Food Hubs perform well in the indicator categories associated with increasing food access, the aggregate score across all Food Hubs in our sample remains relatively low.

Less than 50 percent of the Food Hubs in our sample make a strong contribution to the 'improved food access' indicator category overall.

Our second social sustainability indicator, 'improved food security', seems to receive far less attention than 'improved access to fresh, healthy produce'. The majority of Food Hubs in our sample either place little emphasis on the objective or do not at all mention the objective of improving food security or assisting food insecure populations. This may be related to the location of the Food Hubs in our sample. While some of our Food Hubs are located in the neighborhoods with high poverty rates, 57 percent of our Food Hubs are located in neighborhood with income levels above the national median poverty rate and some are located in high-income areas. Some Food Hubs may make an indirect contribution to improving food security by donating their unsold produce [40]. This may benefit food-insecure households, however, when no explicit mention of such benefits was made, we did not consider it a contribution to improved food security, and there is no evidence that all Food Hubs adopt this practice [77]. It must also be recognized, however, that the economic viability of a Food Hub suffers when it is located in a neighborhood with low purchasing power and a high poverty rate [41]. The low level of contributions to our 'improved food security' indicator may therefore be less of a reflection of the commitment of the Food Hubs and more of a reflection of the economic realities they face. Economic incentives may be needed for Food Hubs that serve food-insecure populations in order to increase the overall contribution of Food Hubs to this social sustainability indicator. Previous studies estimate, for example, that one-third of U.S. Food Hubs highly depend on grant funding to carry out their core functions [75].

With respect to the environmental sustainability contributions of Food Hubs, we find that Food Hubs play a significant role in reducing food transportation. Among the Food Hubs in our sample, 75 percent have large memberships and predominantly procure from local farmers. They also expressly support small- and medium-sized local farms. The local food systems network the Food Hubs facilitate may thus pave the way for a viable substitute for fruit and vegetable importations produced at longer distances. Food Hubs also reduce negative transportation-related externalities on the consumer side. Some Food Hubs aggregate deliveries of produce from various local farms to easily accessible drop-off sites, while others deliver to local supermarkets and corner stores, school cafeterias, restaurants, and caterers. Among the Food Hubs in our sample, 61 percent work with at least two local food distributors to make local food alternatives more viable.

Our Food Hubs also generally encourage their farmers to pursue healthy production methods. Among the Food Hubs in our sample, 25 percent are actively engaged in the production process of their local farms by offering technical assistance and training to their member farmers. Several of the Food Hubs state that their goal is to expand food production practices which seek to improve soil health, lower pesticide use, and adopt other more sustainable production practices. Given the location bias of our Food Hubs in predominantly urban and metropolitan areas, the level of involvement of the Food Hubs on the production side of the local food system is an encouraging finding. Further improvements in this indicator category may result from an explicit networking role that the Food Hubs can play in linking farmers to share their best practices and advancements in sustainable food production.

The food waste reduction contribution of Food Hubs stemming from reduced transportation and storage related losses can be considered substantial even though explicit waste reduction practices are less frequent. We deduce the positive contribution from the fact that almost all of the Food Hubs in our sample mention their commitment to sourcing food locally from a network of local farmers who constitute their Food Hub membership. This means that food not only travels shorter distances but is also less likely to suffer losses along the distribution chain. Moreover, retailers and customers of Food Hubs tend to be more inclined to prioritize locally grown food and are likely to be more tolerant toward so-called 'ugly vegetables' which may be somewhat misshapen and do not meet the common size and color standards that customers expect of packaged food and supermarket

food. In addition, most of the Food Hubs in our sample donate unsold produce, which also serves a social sustainability role, as mentioned previously. This also reduces the food waste stream. However, only six Food Hubs in our sample compost their food waste. We also did not find any evidence of education or training programs focused on food waste reduction, such as composting classes, for example. Given the producer and consumer networks Food Hubs represent, this seems to be a missed opportunity. Added positive contributions to waste reduction could accrue if Food Hubs adopted a more active role in waste reduction, composting, and related matters.

Finally, we find that very few of the Food Hubs in our sample focus on our fourth environmental sustainability indicator, namely 'improved water management'. Promoting water management was largely absent from the documented objectives or program activities of our Food Hubs. This may be at least partly due to gaps in the information available in publicly accessible documents. Considering the motivation of Food Hubs' to present their business models as promoting responsible food production, the lack of explicit attention to more efficient water use practices seems surprising, especially in areas that have experienced water shortages due to droughts. Further research is needed to explore why sustainable water management is not more prominently featured and whether the limited contributions to this indicator imply a lack of communication or a lack of water management contributions [78]. Should the latter be the case, a good starting point may be to add water management as a topic to the technical assistance activities that the Food Hubs provide to farmers in their network. Our aggregate ranking in Figure 4 indicates that, consistently with literature [79], Food Hubs make significant social contributions, notably by increasing access to locally produced fruits and vegetables. However, attention to food insecurity is sparse. Food hubs significantly contribute to environmental sustainability by reducing food transportation, shortening food supply chains, and reducing the time and distance between harvest and consumption [80]. While there is evidence of a growing awareness of the food waste and loss problem of the U.S. food system, explicit waste reduction strategies are sparse among the Food Hubs in our sample suggesting significant room for improvement through efforts such as composting and educating producers and consumers on waste reduction strategies. In light of the severe water shortages in several of the prime specialty crops production areas in the U.S., there also appears to be room for improvement in both implementing water saving methods and educating consumers and producers on improved water management strategies.

*Limitations of the Study*

While our study reveals several interesting observations about the social and environmental sustainability contributions of Food Hubs, it also exposes some limitations. First, the scope of our analysis was limited by our use of publicly available information. Given the range of materials that proved to be publicly available, variations in the quality and accuracy of information presented by the Food Hubs themselves on their websites are necessarily reflected in our data. We addressed these inconsistencies in the available information to some extent by developing our sub-indicator based on prevalent social and environmental sub-categories that were frequently mentioned in the publicly available information (see Tables A1 and A2 of the Appendix A). Secondly, differences in the extent to which the mission statements of the Food Hubs elaborated on their social and environmental sustainability contributions may underscore variations in the degree to which the Food Hub managers themselves care about these contributions or they may reflect larger objectives that prevented a stronger emphasis on these contributions on the website of a Food Hub. A more comprehensive data-gathering effort (e.g., surveys and/or interviews with Food Hub managers) could alleviate these limitations. Finally, content analysis is subject to inferences from text rather than quantitative measures resulting in a potentially lower accuracy of results and limitation in capturing differences. We use quantitative ratings and introduce average achievement scores to quantify the differing levels of social

and environmental contributions of the Food Hubs. Future evaluations of Food Hubs could facilitate a comparison of trends in contributing to sustainability criteria over time.

## 4. Conclusions

We examine the sustainability contributions of 50 Food Hubs across the United States by developing a framework of two social and four environmental indicators. Our findings suggest that Food Hubs strongly contribute to increasing food access, however, there is room for improvement in their contribution to food security. We recognize the tension between improving food security and profitability, especially in low-income areas, where purchasing power and therefore the ability to generate revenue are limited. This suggests the need for policies that can bridge the affordability and sustainability objectives of Food Hubs and the U.S. food system in general. With respect to the environmental aspects of our study, our findings suggest that Food Hubs significantly contribute to reducing long-distance food transportation and associated food losses by sourcing food locally. Although our findings show that deliberate efforts to reduce food waste and loss are limited. We do not find strong support in the public data on which our analysis is based to indicate that Food Hubs play an active role in improving the water management practices of the farmers in their network.

While our results are overall encouraging, they also point to several opportunities for additional research. Our study used publicly available information and analyzed it to answer the question of whether Food Hubs in U.S. cities and metro areas make a positive contribution to social and environmental sustainability. Further research might focus on collecting specific information that could enhance the choice of sustainability indicators providing the yard stick to measure the contributions of the Food Hubs. As questions about the unsustainability and vulnerability of the current U.S. food system persist, our analysis offers a viable starting point for assessing the sustainability contributions of a more localized and decentralized food system. We invite further research on this important topic especially in light of current pressures on energy prices, and the persistent risks of external shocks that may continue to disrupt highly centralized food systems nationally and globally.

**Author Contributions:** Conceptualization, S.O. and H. S.; methodology, S.O. and H.S.; data collection, H.S.; analysis, S.O. and H.S.; original draft preparation, S.O. and H.S.; review and editing, S.O., H.S., R.G., and M.S. All authors have read and agreed to the published version of the manuscript.

**Funding:** The authors acknowledge funding support through the USDA NIFA AES funds of the University of the District of Columbia for conducting this research.

**Data Availability Statement:** Not applicable.

**Acknowledgments:** The authors acknowledge comments and suggestions provided by Maros Ivanic, as well as participants in the Urban Food System Symposium in Kansas City where preliminary findings of this research were presented.

**Conflicts of Interest:** The authors declare no conflict of interest.

## Appendix A

**Table A1.** Social Sustainability Sub-indicators and the Rating Schedule.

| | Rating Scale | | | |
|---|---|---|---|---|
| | **0** | **1** | **2** | **3** |
| **Increasing Access to Fresh and Healthy Food** | | | | |
| Does the Food Hub state a social mission for increasing access to healthy and fresh food for local consumers? Source: Food Hub websites | Not mentioned | Implied but not stated | Stated without a strong emphasis | Stated with a strong emphasis |

| | 0 | 1 | 2 | 3 |
|---|---|---|---|---|
| Does the Food Hub supply food to local K-12 schools, hospitals, or senior care? | No | - | - | Yes |
| Source: Food Hub websites and USDA Food Hub directory | | | | |
| Does the Food Hub offer convenient and effective means for the direct sale of products to consumers? | Based on the number of means of direct sales offered by the Food Hub among a farmers market, online sale, or community-supported agriculture (CSA). | | | |
| Source: USDA Food Hub directory | | | | |
| Does the Food Hub offer training programs to consumers to promote healthy choice and use of food? | No | - | - | Yes |
| Source: Food Hub websites and USDA Food Hub directory | | | | |
| **Addressing Food Insecurity** | | | | |
| Does the Food Hub state a social mission to solve food insecurity in the region where it operates? | Not mentioned | Implied but not stated | Stated without a strong emphasis | Stated and strong emphasis |
| Source: Food Hub websites | | | | |
| Is the Food Hub located within a reasonable distance from a low-income neighborhood? | The poverty rate in the first quartile of national poverty rate distribution | The poverty rate in the second quartile of national poverty rate distribution | The poverty rate in the third quartile of national poverty rate distribution | The poverty rate in the fourth quartile of national poverty rate distribution |
| Source: US Census, Small Area Income and Poverty Estimates (SAIPE) Program | | | | |

**Table A2.** Questions to Evaluate Environmental Sustainability Contributions of Food Hubs.

| | Rating Scale | | | |
|---|---|---|---|---|
| | **0** | **1** | **2** | **3** |
| **Promoting Healthy Food Production** | | | | |
| Does the Food Hub require or highly encourage healthy food production practices (organic production, chemical-free production, or good agricultural practices (GAPs))? | No | - | - | Yes |
| Source: USDA Food Hub directory | | | | |
| Does the Food Hub offer technical support or training programs to farmers about any aspect of food production? | No | - | - | Yes |
| Source: Food Hub websites | | | | |
| **Reducing Food Transportation** | | | | |
| Does the Food Hub have a large membership? | Single farm | 2–9 farms | 10–50 farms | Greater than 50 farms |
| Source: Food Hub websites | | | | |
| Does the Food Hub procure from local farmers? | No | Predominantly from distant farms (>100 miles) | Predominantly local farms | Only local farms |
| Source: Food Hub websites | | | | |
| Does the Food Hub offer home delivery, CSA, or neighborhood drop-off? | No | - | - | Yes |
| Source: Food Hub websites | | | | |
| Does the Food Hub sell products to local food businesses? Number among the following types: local large supermarkets, small grocery stores, corner stores, food retailers, restaurants, caterers, distributers, and food processors. | Fewer than 2 | 2–3 | 4–5 | 6–8 |

| | | | | |
|---|---|---|---|---|
| Source: USDA Food Hub directory | | | | |
| **Reducing Food Waste and Loss** | | | | |
| Does the Food Hub donate unsold products? | No | - | - | Yes |
| Source: Food Hub websites and USDA Food Hub directory | | | | |
| Does the Food Hub actively compost waste produce? | No | - | - | Yes |
| Source: Food Hub websites | | | | |
| **Promoting Water Management** | | | | |
| Does the Food Hub encourage water-saving production? | No | - | - | Yes |
| Source: Food Hub websites | | | | |

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
