# Peer review of "Are Food Hubs Sustainable? An Analysis of Social and Environmental Objectives of U.S. Food Hubs"

_sustainability, doi:10.3390/su15032308_

Round 1
Author Response
Reviewer 1 suggested a better clarification of the paper’s contributions and recommended to inserting this toward the beginning of the paper. We have incorporated this suggestion in the introduction of the paper as shown in the revised draft.
Reviewer 1 also suggested moving these tables to the appendix and citing them in the text. We have done that and have prepared an appendix which makes the text itself more readable. In making these changes we have followed the instructions of the journal.
Reviewer 1 further suggested making the conclusion section more concise and we have now limited the conclusion to two short paragraphs summarizing our main findings.
Reviewer 2 Report
Dear Authors,
Can you please incorporate the following suggestions.
1- Scope of study is not clear please improve it
2- In all the results and measurement mentioned in the tables does not have error values? Ensure the error obtained.
3- Improve the conclusion with few more information.
4-The limitation of the work is not explained. Explain at least half of the page before the conclusion
5- In discussion section the citation of literature is missing in many places
6- No references found form in the year 2022
7- Check the language and grammar of the paper
8- The novelty of work is missing. Add at end of the introduction
9- Why the current area is chosen for the study? Explain in the introduction
Author Response
Reviewers 2 suggested a clearer exposition of the scope and purpose of the paper. We have addressed this suggestion by revising the introduction of the paper. We also modified the last paragraph and added several references to ensure we put our own research more fully into the context of the existing literature as shown in the reviewed draft of the paper.
Reviewer 2 also suggested that we further explain the limitations of our study. We have included this suggestion by adding a section to the paper explaining the limitations of our methodology. We have inserted this section prior to the conclusion of the paper. We also discuss what we have done to mitigate the limitations stemming from our use if publicly available data to develop our scheme to assess the sustainability contributions of the Food Hubs in our sample. data sources are available.
Reviewer 2 further suggested that we add more recent citations from 2022 given the substantial discussion of the sustainability of food systems in the current literature. We have addressed this suggestion in the revised draft of the paper and have added a total of nine new references of recent contributions to the discussion of sustainable food systems and urban food systems. We thank the reviewer for this suggestion.
Reviewer 2 further suggested making the conclusion section more concise and we have now limited the conclusion to two short paragraphs summarizing our main findings.
Reviewer 3 Report
Thank you for choosing me to review your article. I have provided my comments below. In my opinion, the title of the article is too long and needs to be shortened. It is very good that the keywords do not duplicate the words in the title. The purpose of the work should be more emphasized. It is not entirely clear to me why the authors divided the article into parts in this way. Why are the results not in the same subchapter as their discussion, but together with the methodology? Conclusions are correct, but too extensive and should definitely be presented in a more concise form.
Author Response
The reviewer suggested shortening the title and we propose a new title as follows: “Are Food Hubs Sustainable? An Analysis of Social and Environmental Objectives of U.S. Food Hubs”.
Reviewers 3 suggested a clearer exposition of the scope and purpose of the paper and we have addressed this suggestion by revising the introduction of the paper as shown in the reviewed draft. We have also modified the last paragraph and added several references to ensure we put our own research more fully into the context of the existing literature.
Reviewer 3 further suggested making the conclusion section more concise and we have now limited the conclusion to two short paragraphs summarizing our main findings.